# Metagenome Profiling Identifies Potential Biocontrol Agents for *Selaginella kraussiana* in New Zealand

**DOI:** 10.3390/genes10020106

**Published:** 2019-01-31

**Authors:** Zhenhua Dang, Patricia A. McLenachan, Peter J. Lockhart, Nick Waipara, Orhan Er, Christy Reynolds, Dan Blanchon

**Affiliations:** 1Ministry of Education Key Laboratory of Ecology and Resource Use of the Mongolian Plateau & Inner Mongolia Key Laboratory of Grassland Ecology, School of Ecology and Environment, Inner Mongolia University, Hohhot 010021, China; 2Institute of Fundamental Sciences, College of Sciences, Massey University, Palmerston North 4442, New Zealand; p.a.mclenachan@massey.ac.nz (P.A.M.); p.j.lockhart@massey.ac.nz (P.J.L.); 3The New Zealand Institute for Plant & Food Research Limited, Mt Albert, Auckland 1142, New Zealand; nick.waipara@plantandfood.co.nz; 4Arborlab Consultancy Services, Auckland 0632, New Zealand; orhaner8@gmail.com; 5Greenscene New Zealand Ltd., Auckland 1024, New Zealand; christy.reynolds@greenscenenz.com; 6School of Environmental and Animal Sciences, Unitec Institute of Technology, Private Bag 92025, Auckland 1142, New Zealand

**Keywords:** *Selaginella kraussiana*, invasive species, biocontrol agents, metagenome profiling

## Abstract

Metagenomics can be used to identify potential biocontrol agents for invasive species and was used here to identify candidate species for biocontrol of an invasive club moss in New Zealand. Profiles were obtained for *Selaginella kraussiana* collected from nine geographically disjunct locations in Northern New Zealand. These profiles were distinct from those obtained for the exotic club moss *Selaginella moellendorffii* and the native club mosses *Lycopodium deuterodensum* and *Lycopodium volubile* also collected in Northern New Zealand. Fungi and bacteria implicated elsewhere in causing plant disease were identified on plants of *Selaginella* that exhibited signs of necrosis. Most notably, high densities of sequence reads from *Xanthomonas translucens* and *Pseudomonas syringae* were associated with some populations of *Selaginella* but not *Lycopodium*. Since these bacteria are already in use as biocontrol agents elsewhere, further investigation into their potential as biocontrol of *Selaginella* in New Zealand is suggested.

## 1. Introduction

Invasive plants are generally thought to have a competitive advantage over native plants in their new range because they have few or no natural enemies [1,2,3,4]. However, the longer a species has been in the geographical area they have invaded, the more likely it is to accumulate pests and pathogens, reducing that advantage [3,5,6,7]. It can be expected that the wider the range an invasive plant species has, the more likely it is that it will encounter pathogens native to the region that are capable of infecting it [3,6,8]. In addition, the longer an invasive species has been in a geographical region, the more likely a pathogen from its original home range will also be present [6]. Other factors, such as degree of relatedness between invasive and native plant species [9], diversity of native pest and pathogen species, and resource availability [10], can all affect pathogen accumulation. The diversity of pathogens that an invasive plant exhibits will thus depend on the extent of elimination of co-evolved pathogens during the invasion of its new habitat and the subsequent gain of pathogens native to its new range [11]. Potential biological control agents can be sourced from the original home range of the invasive plant [12,13,14,15] or from the new range by screening diseased material for candidate fungi, bacteria, and viruses [3,9,16]. As we illustrate here, metagenome profiling using high throughput sequencing can be used to screen plants exhibiting symptoms of disease to identify candidate organisms for biocontrol of invasive species.

*Selaginella kraussiana* (African club moss) is an invasive club moss species that has naturalised in Australasia, Europe, and North, Central, and South America [17]. In New Zealand, it has subsequently established in shady damp conditions in lowland bush areas, home gardens, and waterways throughout most of the country. The species forms thick forest floor mats that can reduce species richness and suppress seedling growth of native plants [14,18]. *Selaginella kraussiana* is potentially a useful organism to observe the interplay between pathogen release and pathogen accumulation. In New Zealand, it was first reported in Wellington and the Bay of Islands in 1919 [19] and has subsequently established throughout the country, so it has a wide geographic spread and has had nearly a century of time for native pathogens to infect the species. The closest native relatives (and potential source of pathogens) of *S. kraussiana* (Selaginellaceae) are the native club mosses in the genera *Lycopodium*, *Lycopodiella*, and *Huperzia* (Lycopodiaceae). As yet there appears to be little knowledge of naturally occurring pathogens of *S. kraussiana* in New Zealand or elsewhere [20]. Some fungal species with varying degrees of pathogenicity towards the species have been reported recently in collections of *S. kraussiana* in New Zealand [21,22], and more recently a study included a comparison with the native club mosses *Lycopodium deuterodensum* and *Lycopodium volubile.* Here, we analyse the microbial profiles of 18 samples of *S. kraussiana* collected from nine disjunct locations in New Zealand matched against protein databases to gain a better understanding of the plant–fungal and plant–bacteria associations. Microbial profiles were also determined for accessions of the related invasive plant *Selaginella moellendorffii* and native *L. deuterodensum* to aid as references.

## 2. Materials and Methods 

### 2.1. Sample Collection

Field collection involved sampling from nine sites in the Auckland, Northland, and Waikato regions (Figure 1). At each site, plant material was collected from two spatially separated subsites. Samples all came from a single plant at each subsite. Subsites were chosen where *S. kraussiana* showed signs of dieback, yellowing or other discolouration (Figure 2) and were also far enough removed from the track to avoid human or animal factors in plant symptoms. Leaves, stems, and strobili were collected from each subsite, and samples were selected to include the junction between green and discoloured tissue. Material was collected using instruments sterilized in 70% ethanol, soil and roots were avoided, and plant material was placed in a sealed plastic bag with a damp cloth to prevent drying out. A hundred milligrams of material showing signs of disease was placed into 1 mL of RNA*later* (Ambion, Austin, TX, USA) in a sterile 1.7 mL microtube. The samples were stored at −20 °C prior to DNA extraction and DNA sequencing.

### 2.2. DNA Extraction

DNA was extracted from the samples using the Nucleospin Plant II kit from Macherey–Nagel (Düren, Germany) and the protocol for Genomic DNA from Fungi outlined in the user manual. The manufacturer’s protocol was followed except for the following modifications: The plant material was removed from the RNA*later* into a screw cap micro tube (Sarstedt, Nümbrecht, Germany). Twenty to thirty 2-mm Zirconia Beads (11079124zx; BioSpec Products, Bartlesville, OK, USA) were added to each sample, along with 300 µL of Buffer PL1 (from the kit), and the screw cap was sealed with parafilm. The samples were homogenized with the beads in a Roche MagnaLyser (Mannheim, Germany) for 90 s at 5000 rpm, RNase was not added, and the samples were incubated for 20 min at 65 °C. The protocol was followed exactly after addition of the chloroform and the DNA was eluted in 50 µL of Elution Buffer. The concentration of DNA ranged from 0.7–17.9 ng/µL, and the purity as measured on a Nanodrop as a 260/280 ratio varied between 1.37 and 2.07, with most samples between 1.7 and 1.9.

### 2.3. Sequencing

Nextera XT libraries were prepared for the Illumina MiSeq. They were prepared manually following the manufacturer’s protocol (15031942; Illumina, San Diego, CA, USA). DNA extracts were quantified and normalized to 0.2 ng/µL using a Quant-iT PicoGreen assay system (Q33120; Thermo Fisher Scientific, Waltham, MA, USA) on a Qubit 3 reader (Thermo Fisher Scientific) and then fragmented and tagged via tagmentation for 8 minutes at 55 °C (default is 5 minutes) to ensure optimal fragmentation by the enzyme “transposase”. Prior to amplification, 5 µL of stop tagmentation buffer (NTA) was added to stop any enzymatic activity. PCR enrichment was performed by adding 30 µL of amplification master mix to Illumina primers and incubated using the following protocol: 72 °C for 3 minutes, 95 °C for 30 seconds, 15 cycles of 95 °C for 10 seconds, 55 °C for 30 seconds and 72 °C for 30 seconds, 72 °C for 5 minutes and, lastly, a hold at 10 °C. It was followed immediately by an AMPure XP bead cleanup (A63880l; Beckman Coulter, Takanini, Auckland, New Zealand) using PEG: A NaCI ratio of 0.6x to retain more than 450 bp products. The Nextera XT libraries were then further quantified for fragment size and DNA concentration using the LabChip GX Touch high sensitivity 3k Assay (Perkin Elmer, Melbourne, Australia) and Qubit 3.0 fluorometer (Thermo Fisher Scientific). The Nextera XT libraries (sample 5734) were then diluted to 2 nM in pooling buffer (10 mM Tris-HCL pH 8.0, dH_2_O, and 0.1% (v/v) of Tween-20), prior to sequencing library cluster generation and sequencing on a MiSeq instrument. 

### 2.4. Data Processing

Illumina MiSeq 2x150 base pair reads were firstly processed for quality using a standard Illumina sequence analysis pipeline. Quality trimming was performed using DynamicTrim (a program of the SolexaQA [23]) with an error probability cutoff of 0.01 to remove any low-quality base calls. Follow this processing, the second round of data trimming was executed using Trimmomatic (version 0.32 [24]). Reads were hard trimmed to 120 bp by executing the Trimmomatic commands MINLEN:120 and CROP:120. The 120 base pair reads were then matched with default parameters against a local version of the NCBI nr database (nr_2015_05_05) using DIAMOND (version 0.7.9 [25]). The weighted LCA algorithm in MEGAN CE (version 6.9.2 [26]) was used in pair-end mode to make taxonomic assignments. To validate the taxonomic assignments, we also performed taxonomic classification using Kaiju (version 1.5.0 [27]) by submitting our data to the Kaiju Web Server http://kaiju.binf.ku.dk/server. In these analyses, the NCBI BLAST nr + euk database (updated 16 May 2017) was used. We chose the Greedy model and default parameter settings. The classification results obtained from MEGAN and Kaiju were visualized in Microsoft Excel and compared using a Wilcoxon signed-rank test. Note that as the number of paired-end reads with quality Phred quality score 30 varied per sample, the number of paired-end taxonomic assignments with MEGAN–LCA also differed per sample. With MEGAN, we normalized counts to 186,029 paired-end assignments per sample. To examine whether data base representation might have impacted inferences made concerning the presence and absence of fungal pathogens, paired reads used in DIAMOND–MEGAN and Kaiju analyses were independently mapped using BWA [28] (default parameter settings) to the internal transcribed spacer (ITS) regions of *Phoma selaginellicola* and *Pestalotiopsis clavispora* which had previously been cultured from *Selaginella kraussiana* in New Zealand [21,22]. The reads were also mapped to the ITS region of *Parastagonospora phoenicicola* and *Parastagonospora caricis*, since this genus was identified in DIAMOND–MEGAN analyses of *S. kraussiana.* Of interest was determining whether analyses of ITS sequences would suggest the presence of *Phoma* and *Pestalotiopsis* on any of the accessions *S. kraussiana,* since these fungal species had previously been cultured from *S. kraussiana* in New Zealand [21,22].

## 3. Results

Figure 1 shows a principle coordinates analysis (PCoA) plot visualizing Bray Curtis dissimilarities in the DIAMOND–MEGAN bacterial and fungal taxonomic profiles of 18 samples of *S. kraussiana*, two samples of *S. moellendorffii*, and two samples of *Lycopodium*. Appendix A provides details on taxonomy assignments for all samples made with MEGAN–LCA analyses. The *S. kraussiana* profiles from the nine disjunct locations tended to be very similar and distinct from those of the other club moss species. Figure 3 shows the relative number of paired sequence reads in each of the 18 samples of *S. kraussiana* that matched to bacterial and fungal genera implicated in plant disease elsewhere. While the microbial profiles on *S. kraussiana* were relatively similar between locations, Figure 3 shows significant variation in the frequencies of reads assigned to genera represented by potential pathogens. Microbial profiling using DIAMOND database matching and taxonomic assignment with MEGAN–LCA identified four genera of bacteria listed among the top 10 pathogenic bacteria of plants [29]. These were *Agrobacterium*, *Erwinia*, *Pseudomonas*, and *Xanthomonas.* The same bioinformatic analyses also identified four genera of fungi listed among the top 10 pathogenic fungi of plants [29]. They were *Botrytis*, *Magnaporthe*, *Colletotrichum*, *Fusarium*, *Ustilago*, *Melampsora*, and *Puccinia*. Other fungal genera identified representing putative pathogens were *Dothistroma*, *Pseudocercospora*, *Sphaerulina*, *Zymoseptoria*, *Aureobasidium*, *Leptosphaeria*, *Parastagonospora*, *Bipolaris*, *Pyrenophora*, *Setosphaeria*, *Verticillium*, and *Rhizoctonia*. 

Figure 4 visualizes the relative number of MEGAN–LCA assignments to bacterial and fungal species implicated elsewhere in plant pathogenesis. Appendix A details the read counts used to construct Figure 4. Appendix A provides results from comparative analyses with Kaiju. Very similar trends were observed between Kaiju and DIAMOND–MEGAN analyses in the relative number of taxonomic assignments made to bacterial and fungal taxa. A relatively high number of reads were assigned to *Pseudomonas syringae* [29] at 12 subsites, including City Walk Track (CW5-1and CW5-3), Earnies Track, Hunua Ranges (HU9-2), Otaika Valley, Wangarei (OT2-1, OT2-2 and OT2-3), Spragg Bush, Waitakere Ranges (SB3-1and SB3-2), Tramline Track, Waitakere Ranges (TL4-1and TL4-2), Waiheke Forest (WI7-1), and Whareora Rd, Wangarei (WH1-3). *Xanthomonas translucens* [30] was identified at the subsites SB3-1, SB3-2, TL4-1, and WH1-3 and was most common at WH1-3. *Xanthomonas vesicatoria* [31] was identified at 9 subsites (CW5-1, CW5-2, CW5-3, OT2-2, SB3-1, SB3-3, TL4-1, WH1-1, and WH1-2). A smaller number of reads assigned to *Agrobacterium rubi*, *Agrobacterium tumefaciens*, *Agrobacterium larrymoorei* and *Agrobacterium vitis* [32] occurred at almost all subsites. All the above species are known to be pathogenic on different host plants under certain conditions. Figure 4 shows that in terms of potential fungal pathogens, most reads were assigned to the genera *Pyrenophora* and *Parastagonospora* (reads number > 2000). While not in the list of the top 10 fungal pathogens [33], *Pyrenophora teres* [34] and *Pyrenophora tritici-repenti* [35] are pathogens of barley and wheat, and these species predominated at almost all of the subsites. *Parastagonospora nodorum* [36], a pathogen of wheat, was also identified at most subsites (all except for SB3-1 and SB3-2).

BWA mapping of the paired reads from *S. kraussiana* to ITS sequences from *Parastagonospora* (*P. phoenicicola* and *P. caricis*) produced a low number of matches: 1–77 reads per subsite across all populations. A lower number of the same read set mapped to *Pestalotiopsis clavispora* (1–8 reads). Greater variation for the same read set mapped to *Phoma selaginellicola* (1–257 reads). As discussed below, the low read number contributing to limited phylogenetic resolution of the sequenced ITS regions makes interpretation of these observations problematic.

## 4. Discussion

Metagenomic profiling was conducted using DIAMOND–MEGAN–LCA short read sequence analyses, and the findings corroborated using Kaiju. Both methods matched DNA sequence reads against protein databases. Numerous fungal and bacterial species were identified that are known plant pathogens. Potentially pathogenic *Xanthomonas* and *Pseudomonas* species occurred at high frequency on some populations of *Selaginella*. Similarly, *Pyrenophora, Bipolaris*, and *Parastagonospora* were relatively common pathogenic fungi also occurring on *Selaginella*. A significant level of variation in relative read numbers for potential pathogens occurred between samples (Figure 3). This might in part be explained by different extents of disease progression at different sites or in different stages of disease progression of the sampled tissue. Differences in the pathogen dynamics resulting in disease progression could also occur at the different sites. More intensive sampling, not possible with this initial survey, would be required to evaluate these possibilities, and a weakness of the present study is the unknown relationship between the appearance of necrotic symptoms on the plants and the state of disease.

Nevertheless. despite this shortcoming, it is interesting that *Xanthomonas* and *Pseudomonas* are bacterial genera under consideration as bioherbicides for weeds elsewhere [16], and the large number of reads associated with some diseased samples encourages investigation of these genera and the species identified as potential biological control agents for *Selaginella* in New Zealand. In further evaluating their biocontrol potential, future studies should include sampling of necrotic and visibly healthy tissues at sites where *S. krausianna* is found, as well as more intensive sampling of *Lycopodium Lycopodiella* and *Huperzia* species, as their sampling was very limited in the present study.

As both DIAMOND–MEGAN–LCA and Kaiju analyses involved database matching of translated reads to protein databases, future work to complement our findings in the present study could involve matching the sequenced reads to fungal ITS and 18S rDNA databases. These markers do not produce the phylogenetic resolution of whole genome shotgun sequences, but there is currently much greater phylogenetic representation of diverse fungal phylogenetic lineages for rDNA loci than for coding regions and genomes [37,38]. Neither MEGAN nor Kaiju analyses identified *Pestalotiopsis clavispora* or *Phoma selaginellicola* on any of the plants surveyed. These fungal species had previously been cultured from New Zealand *Selaginella kraussiana* and were identified based on ITS sequence analyses [21,22]. Genome information from both species was poorly represented in the databases used by Kaiju and MEGAN in the present study, and thus it is possible that lack of representation for these and possibly other fungal species in our results might be explained by poor database representation. This question was investigated in the present work by mapping the paired end reads from the *S. kraussiana* samples to ITS sequences from two fungal species previously isolated on *Selaginella* in New Zealand, and also mapping the same read sets to the ITS region of a fungal species that was identified by our protein-based homology assignments in the present study. Overall, the number of reads mapping to all reference sequences was very low, and this finding cautions against over interpretation of our mapping results with BWA. A more thorough investigation of this issue is warranted, and this will require specifically targeting rDNA loci during Illumina library preparation to ensure even and appropriate coverage of marker regions for statistical analyses.

## Figures and Tables

**Figure 1 genes-10-00106-f001:**
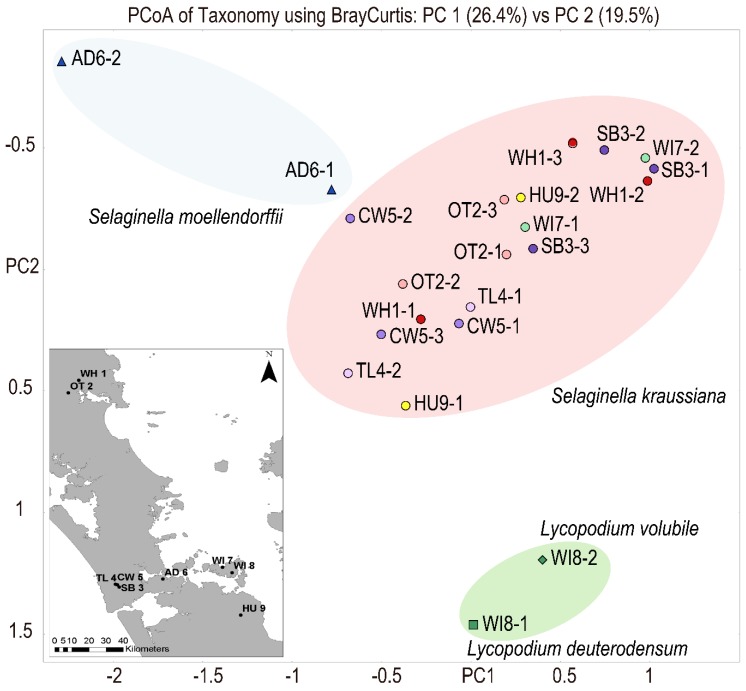
Sampling locations and principal coordinates analysis (PCoA) of Bray Curtis dissimilarities in taxonomic content of club moss species. Blue triangles indicate two *Selaginella moellendorffii* samples. Green diamond and square represent *Lycopodium volubile* and *Lycopodium deuterodensum*, respectively. Filled Circles indicate *Selaginella kraussiana* samples. Sites are colour-coded. Location codes used are those from an earlier report prepared for the Auckland Council WH1: Whareora Rd, Whangarei; OT2: Otaika Valley, Whangarei; SB3: Spragg Bush, Waitakere Ranges; TL4: Tramline Track, Waitakere Ranges; CW5: City Walk Track, Waitakere Ranges; AD6: Fernz Fernery, Auckland Domain; W17: Waiheke Forest; W18: Te Matuku Bay; HU9: Earnies Track, Hunua Ranges. Subsites within each location are denoted 1–3.

**Figure 2 genes-10-00106-f002:**
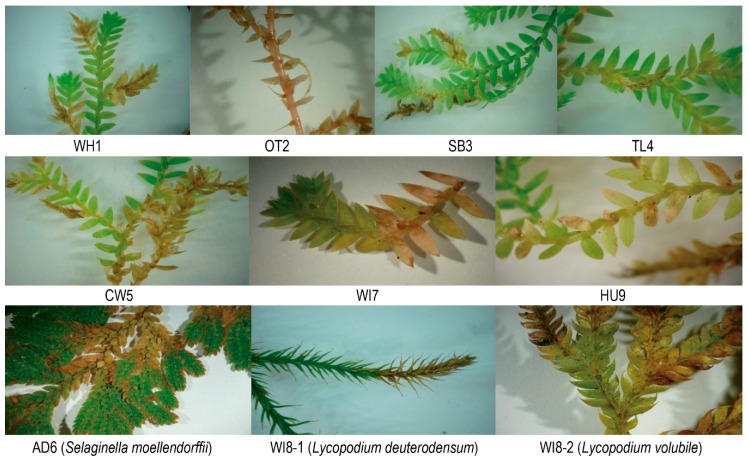
Symptoms suggesting disease on *Selaginella* and *Lycopodium* at nine locations. All *Selaginella* populations exhibited some symptoms of diseased tissue.

**Figure 3 genes-10-00106-f003:**
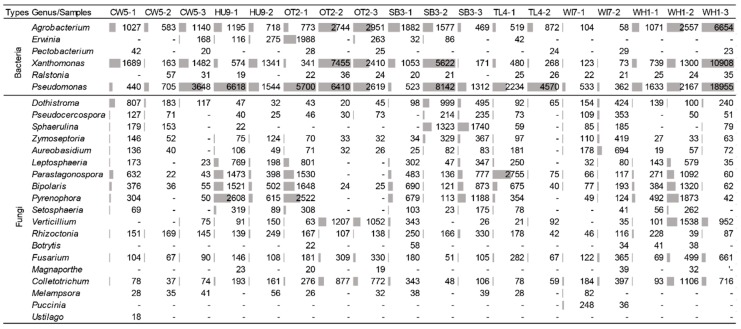
Genera representing potentially pathogenic bacteria and fungi on *S. kraussiana* exhibiting signs of disease at nine locations.

**Figure 4 genes-10-00106-f004:**
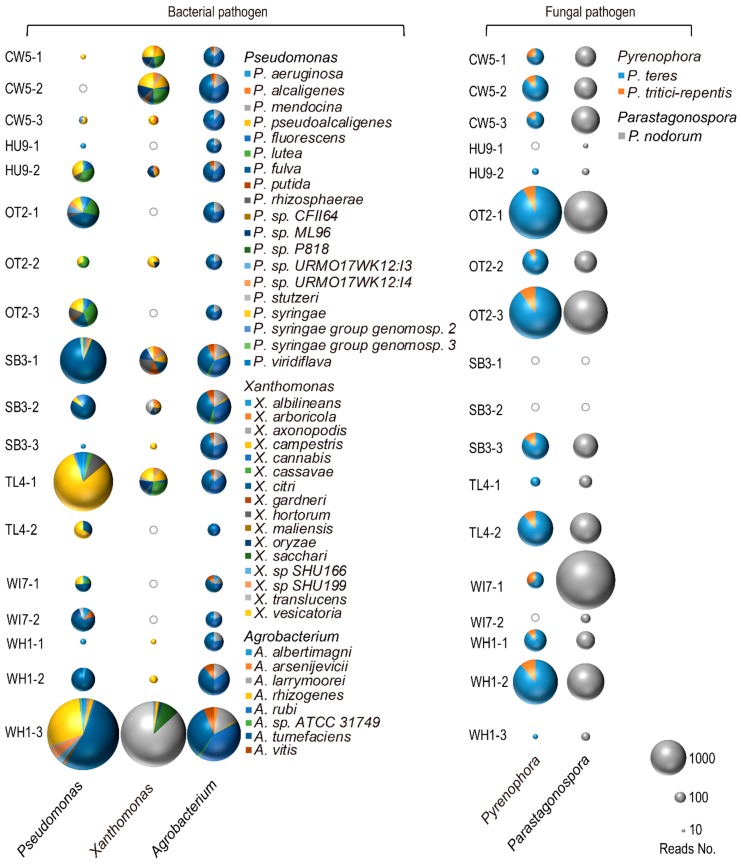
Three-dimensional bubble-pie plot showing species of potentially pathogenic bacteria and fungi on *S. kraussiana* at seven locations. The relative number of reads assigned to each species is indicated by the size and area of each bubble. Grey hollow circles indicate where no reads were assigned to the relevant subsite.

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
