# Peer review of "Metagenome Profiling Identifies Potential Biocontrol Agents for Selaginella kraussiana in New Zealand"

_genes, 2019, doi:10.3390/genes10020106_

Reviewer 1 Report

This paper discusses the application of metagenomic sequencing to both invasive and native mosses in New Zealand so as to determine possible biocontrol agents to combat the invasive mosses. Protein-alignment based analyses of the metagenomic sequencing reads is performed using both DIAMON+MEGAN and Kaiju. Elevated levels of Xanthomonas translucens and Pseudomonas syringae were found associated with the invasive mosses, but not in the native mosses. The authors suggest that these bacteria merit further consideration as biocontrol agents.

This study nicely demonstrates how a straight-forward metagenomic analysis can provide a useful insight into microbial associations with host species of interest.

Figure 2 shows plants with disease symptoms, which ones are theses in Figure 1 and Figure 4?

The legends for Figures 1 and  Figure 4 need to be improved. What do the colors and different shapes (circles, triangles, squares) indicate in Figure 1? It would be very useful to change the cryptic sample names into something more user friendly, using a prefix to indicate which moss is associated with the sample, e.g. Sm-AD6-2. In Figure 4, the prefixes would help to determine which mosses the samples are from, but in addition, this could also be indicated on side of the figure.

Is Figure 4 really a bubble plot? The bubbles look like pie charts.

On page 2, you repeatedly write (remove once):

In New Zealand it was first reported in Wellington and the Bay of Islands in 1919 [18]

In the discussion you mention analyzing the 16S and 18S reads contained in the sample.

Please consider doing it and reporting on the differences between that analysis and the protein-alignment based analyses.

Reviewer 2 Report

The paper by Dang et al. use metagenomics to identify candidate species for biocontrol of an invasive clubmoss Selaginella kraussianaa in New Zealand.

The paper is very interesting, but it could be very much improved in several aspects. Most importantly about the comparison between Selaginella and Lycopodium (on lines 183-5) the authors wrote “Of particular interest was the finding that potentially pathogenic Xanthomonas and Pseudomonas species occurred at high frequency on some populations of Selaginella but not Lycopodium”. This observation is drawn from the comparison of 9 sites for Selaginella against only two sub-sites both on the Waiheke Island. Focusing only on the Waiheke sites, Selaginella does not seems to show high frequencies of any pathogens and probably it is not been different from Lycopodium. Therefore, as stated later in the discussion a more extensive sampling the other species should be conducted before drawn any conclusion from the comparison between invasive and native clubmosses.

The presentation of the results could be improved. The authors have been very parsimonious in describing their results. Apart from a very brief mentioning of what figure 3 and supplementary material contains (a title) there is no description of which results they contains in the text. In particular, the authors mentioned that they compared the results of MEGAN-LCA and Kaiju and apart presenting two tables in the supplementary material, which by the way they were not accessible, there are no description of these comparisons.

In the beginning of the results are listed all the figures and tables presented in the ms. I think it would be nicer to present them in the paragraph when relevant for the description of the results. Figure 2 is more appropriate in the methods in the Sample collection section. Samples at each sub-site are a single plant or a pool of several plants?

Minor concern

In the abstract line 22: Selaginella and Lycopodium are vascular plant not mosses. If you are referring to something else, please report the species.

Biocontrol and bio-control: choose one and stay with it

Introduction: line 53 and line 58 are repetition.

Line 71: here appear S.moellendorffii. The species is also present in Fig. 1 but not other mention is done in the results.

Line 105 PCR enrichment

Figure 1: h1-1 instead of WH1-1 and there are two "CW5-1".

Line 168: the list of the top 10

Fig.4 a scale for the size of the bubbles would improve the figure

Line 190: Lycopodium,

Author Response

see attached file

Round  2

Reviewer 2 Report

The paper by Dang et al. has improved from the first submission and I willing to suggest its acceptance however in my opinion the comparison between Selaginella and Lycopodium should be taken off because the lack of pathogenic bacteria and fungi could be due simply to the geographical location of those populations considering that the populations of Selaginella in that island showed the same amount of Xanthomonas, Pseudomonas and fungi as Lycopodium. Therefore, I still found your statements

Of particular interest was the finding that potentially pathogenic Xanthomonas and Pseudomonas species occurred at high frequency on some populations of Selaginella but not Lycopodium” very misleading.

You cannot exclude with just two populations that also Lycopodium is affected as Selaginella in other populations.

Minor errors

Line 166 I count 7 genera of fungi in line 167

Line 209 is Pyrenophora

Author Response

Thank you for the further comments. There were 3 points  to address

1) The sentence “Of particular interest was the finding that potentially pathogenic Xanthomonas and Pseudomonasspecies occurred at high frequency on some populations of Selaginella but not Lycopodium” very misleading.

 This sentence has been changed to:

"Potentially pathogenic Xanthomonas and Pseudomonas species occurred at high frequency on some populations of Selaginella."

This change makes the sentence more consistent with our statement in the discussion cautioning  the limitation of the Lycopodium sampling. (line 201 in the attached ms. This version of the ms has all previous track changes accepted and new minor changes shown as track changes)

2) Line 166 I count 7 genera of fungi in line 167  We were not certain of what the reviewer meant?

3) Line 209 is Pyrenophora  - spelling was corrected
